# Prompt, Predict, Correct: LLM-TrajEcho for Closed-Loop Trajectory Forecasting via Online Prompt Feedback

## Abstract

Accurate trajectory prediction is fundamental to the safety of autonomous vehicles. However, state-of-the-art methods often rely on computationally intensive multi-sensor fusion to achieve high precision, which increases system complexity and hinders real-time deployment. Furthermore, most predictors operate in an open-loop manner, suffering from uncorrected error accumulation. In response, we propose LLM-TrajEcho, a lightweight, end-to-end vision-based framework that eliminates the need for sensor fusion while enabling real-time performance and closed-loop correction. Our framework efficiently encodes spatiotemporal features from video sequences and translates them into structured natural language prompts for a large language model (LLM), leveraging Parameter-Efficient Fine-Tuning (LoRA) to ensure computational efficiency. A key innovation is our online closed-loop feedback mechanism, which dynamically refines the LLM's context based on prediction errors, mitigating long-term drift. Experiments on nuScenes and KITTI Tracking datasets demonstrate that LLM-TrajEcho runs at 0.53 ms per sample, achieves competitive ADE, significantly improves FDE by 30%, and MR by 21%. Our work shows that vision-based prediction, combined with LLM reasoning and closed-loop learning, offers a viable path toward accurate and efficient autonomous driving. Demo:`/r/ICLR-1693-Demo/`

## 1 Introduction

The rapid development of autonomous driving technologies (Huang et al., 2025; Cheng et al., 2024) has raised the demand for accurate trajectory prediction, crucial for real-time planning and safe decision-making (Hu et al., 2023b; Wu et al., 2024). Early physical (Wolff et al., 2016) and statistical models (Vasquez & Fraichard, 2004) performed well in simple scenes but struggled in complex environments with diverse motion and uncertainty. Deep learning models such as RNNs (Salzmann et al., 2020; Lee et al., 2017) and LSTMs (Alahi et al., 2016; Sadeghian et al., 2019; Greff et al., 2016) partially addressed these issues, yet suffered from vanishing gradients and limited long-term modeling. Transformer-based approaches (Liu et al., 2024; Jia et al., 2023a; Yan et al., 2023) improved global dependency capture, but their quadratic complexity limits real-time use. Recent efforts reduce this cost via factorized attention (Zhou et al., 2023) and state-space models (Gu & Dao, 2023; Gu et al., 2022; Smith et al., 2024), as seen in Trajectory Mamba (Huang et al., 2025) and DeMo (Zhang et al., 2024). However, these numerical sequence models still lack semantic reasoning, intent modeling, and integration of high-level context, such as interaction logic and traffic rules. This gap hinders both accuracy and efficiency in complex, real-world driving scenarios.

With the widespread adoption of onboard cameras and surround-view systems in mass-produced vehicles, video has become a rich input modality for contextual information. Recent works have explored end-to-end trajectory prediction directly from visual streams, which improves the perception of dynamic agents such as vehicles, pedestrians, and cyclists (Chen et al., 2023b; Moon et al., 2024). At the same time, inspired by the success of large language models (LLMs) in natural language reasoning, several studies have investigated "naturalizing" numerical data, reformulating trajectory prediction as a sequence-to-sequence language modeling task (Bae et al., 2024; Mao et al., 2023). These approaches typically convert historical trajectories into text prompts and let pre-trained LLMs generate future predictions. Other works focus on prompt engineering, where carefully de-

signed instructions are used to guide forecasting (Kwon et al., 2024; Sha et al., 2023). More recently, large-scale vision-language-trajectory frameworks have emerged, combining visual perception and natural language for end-to-end driving tasks (Rowe et al., 2025; Wang et al., 2025).

Despite these advances, three challenges remain open in autonomous driving: (1.) **Video to trajectory data.** While visual encoders can capture rich scene context, effectively transforming motion cues from consecutive frames into structured trajectory representations is still nontrivial. Accurate yet compact input forms are essential for improving downstream decision and planning quality. (2.) **Language as a unified motion state representation.** Some methods leverage text during training but fall back to purely visual or numerical models at inference (Moon et al., 2024). Others rely on manually crafted prompts that lack temporal grounding (Peng et al., 2025). Although there are early attempts to represent trajectories or planning outputs directly as language tokens (Bae et al., 2024; Mao et al., 2023), natural language has not yet been established as a persistent, primary modality across the entire end-to-end pipeline. (3.) **Online closed-loop feedback.** Current LLM-based methods generally remain static during inference. Approaches such as autoregressive decoding or distillation still require offline gradient updates (Lan et al., 2024), which limits their applicability in dynamic and real-time driving environments.

Our work is motivated by the aforementioned challenges. We propose LLM-TrajEcho. To the best of our knowledge, LLM-TrajEcho represents **one of the first** lightweight, end-to-end frameworks that establishes a **fully closed loop** between training and inference for trajectory prediction, particularly within the emerging paradigm of LLM-based motion forecasting. The framework integrates visual feature extraction, motion parsing, natural language prompting, LLM-based inference, and online feedback into a unified pipeline. Specifically, sequential video frames are encoded by a pretrained vision encoder. To further balance accuracy and efficiency, we design a lightweight patch-wise self-attention encoder tailored for motion feature extraction. Following the success of patch token attention (Dosovitskiy et al., 2021a; Liu et al., 2021b) that reduces quadratic pixel-level attention to linear complexity in the number of patches, enabling global dependency modeling under real-time constraints. Extracted features are temporally smoothed to obtain frame-wise displacement estimates, which are then transformed into structured natural language descriptions. These descriptions, combined with task instructions and retrieved contextual examples, form the input prompt. We adopt parameter-efficient LoRA fine-tuning (Hu et al., 2022) on LLMs, enabling scalable context-aware reasoning while maintaining efficiency. The generated textual trajectories are parsed back into numerical coordinates for evaluation. To improve adaptability, we introduce an online feedback mechanism: prediction errors are converted into new context examples and stored in an experience pool. During inference, the context is dynamically updated using similarity-based retrieval and reward signals. This closed-loop design enables parameter-free online adaptation, allowing the model to refine its predictions continuously and mitigate error accumulation in dynamic environments. The contributions of this work are threefold:

- We present **LLM-TrajEcho**, an end-to-end trajectory prediction framework that unifies visual motion parsing, natural language prompting, and LLM reasoning into a single pipeline for accurate and efficient forecasting.

- We introduce a structured prompt design that encodes motion information from video into natural language, fully leveraging the contextual reasoning capacity of large language models beyond conventional numerical or handcrafted prompts.

- We propose an online feedback mechanism that updates in-context examples based on prediction errors and state similarity, enabling gradient-free online adaptation and improving robustness in dynamically evolving traffic environments.

## 2 RELATED WORK

### 2.1 VISUAL FEATURE EXTRACTION AND MOTION INFORMATION PARSING

Recent advances in deep neural networks have promoted the use of pre-trained visual encoders for video analysis, extending beyond static feature extraction to temporal dynamics modeling. Models like ResNet (Targ et al., 2016), ViT (Dosovitskiy et al., 2021b), and CLIP (Radford et al., 2021) have improved feature extraction, self-attention, and cross-modal alignment, respectively. In the video domain, Vision Transformer (Dosovitskiy et al., 2021a) and Swin Transformer (Liu et al., 2021b) ex-

tend patch-based attention to spatiotemporal modeling, and driving-specific encoders such as UniAD (Hu et al., 2023a) and MTR++ (Shi et al., 2024) demonstrate end-to-end trajectory forecasting from videos. However, frame-level features are often insensitive to agent motion, and detection-tracking pipelines incur high latency in dense traffic. Lightweight temporal models also face challenges in noise suppression and scene generalization. To address these issues, LLM-TrajEcho introduces a patch-wise self-attention encoder for motion feature extraction, which reduces attention complexity from quadratic to linear and ensures real-time performance.

## 2.2 NATURAL LANGUAGE– BASED DATA MODELING AND LARGE LANGUAGE MODEL REASONING

In recent years, large language models (LLMs) have achieved remarkable progress in natural language understanding and generation. Their strong capabilities in contextual learning and reasoning have led to outstanding performance across a wide range of tasks. Chain-of-Thought (CoT) (Wei et al., 2023) prompting guides models through explicit step-by-step reasoning, while Program of Thoughts (PoT) (Chen et al., 2023a) explores strategies to convert structured information into program-like natural language instructions, enabling LLMs to perform numerical computations and logical inferences. These approaches have shown promising results in domains such as financial forecasting and medical diagnosis, demonstrating that language-based transformations can effectively leverage the semantic knowledge embedded in LLMs to simplify tasks and enhance reasoning efficiency. However, in the context of trajectory prediction, the challenge of converting continuous motion data into textual descriptions that are both concise and semantically meaningful remains a relatively unexplored area. Existing methods (Peng et al., 2025; Lan et al., 2024; Mao et al., 2023) typically rely on large-scale feature fusion networks or complex multimodal regression models. These approaches often involve a substantial number of parameters and incur high training costs, making them difficult to deploy in real-time systems.

## 2.3 ONLINE FEEDBACK AND DYNAMIC CONTEXTUAL EXAMPLE UPDATE MECHANISM

In autonomous driving, continuously evolving traffic conditions require models to adapt in real time to prevent error accumulation. Some prior works (Hoi et al., 2018; Saadatnejad et al., 2024) employ online learning by updating model parameters during inference, but real-time backpropagation incurs high computational costs. Incremental training schemes (Zhao et al., 2024; Hyder et al., 2022) partially address static model limitations but risk overfitting and increased latency due to frequent updates. Reinforcement learning–based strategies (Zhang et al., 2025) use reward signals for dynamic adjustment but require large interaction datasets. More recently, memory- and example-based online adaptation (Yao et al., 2024a) explore retrieving relevant experiences from an external pool, avoiding gradient updates while enabling continual adaptation. However, such strategies remain underexplored in trajectory prediction, where motion states are complex and feedback signals are noisy. Building upon this direction, LLM-TrajEcho introduces a reward-driven example retrieval mechanism that dynamically updates contextual prompts during inference, achieving parameter-free online adaptation specifically tailored for real-time trajectory forecasting

## 3 PRELIMINARIES

Let the input RGB video sequence be denoted as $X \in \mathbb{R}^{T \times H \times W \times 3}$, where $T$ is the number of frames and $H \times W$ denotes the image resolution. The model first splits the video into individual frames $\mathbf{X}_t, t = 1, \ldots T$, and applies a pretrained visual encoder $\mathcal{E}_v(\cdot)$ to extract high-dimensional visual features from each frame, resulting in: $\mathbf{f}_v(t) = \mathcal{E}_v(\mathbf{X}_t)$, $\mathbf{f}_v(t) \in \mathbb{R}^{D_v}$, where $D_v$ is the dimension of frame-level feature. To capture the motion of the target across consecutive frames, the model introduces a motion parsing function $\mathcal{M}(\cdot)$, which maps the frame-level feature sequence to trajectory data $\mathbf{S}$ [1] at each time step:

$$\begin{aligned} \mathbf{S} &= \mathcal{M}\left(\{\mathbf{f}_v(1), \mathbf{f}_v(2), \ldots, \mathbf{f}_v(T)\}\right), \\ &= \{(x_1, y_1), (x_2, y_2), \ldots, (x_T, y_T)\}. \end{aligned} \tag{1}$$

---

[1] 2D positional displacements

Theoretically, we assume that motion between consecutive frames follows a certain degree of smoothness (Cao et al., 2022; Wang et al., 2022). This assumption allows the use of linear regression or lightweight recurrent models to fit the sequence of visual features and estimate the target's positional changes with high precision. Next, the structured $\mathbf{S}$ is transformed into a textual description using a natural language conversion function $\mathcal{F}(\cdot)$:

$$T_S = \mathcal{F}(\mathbf{S}) = \text{``From } t_1 \text{ to } t_T, \text{ the target moves through positions} \\ (x_1, y_1) \rightarrow \cdots \rightarrow (x_T, y_T).\text{''} \tag{2}$$

The final prompt is constructed by combining the predefined task description $H$, the trajectory text $T_S$, and the contextual examples $C$: Prompt $= H \oplus T_S \oplus C$, where $\oplus$ denotes the text concatenation operation. A fine-tuned large language model $\mathcal{E}_{LLM}(\cdot)$ then leverages its contextual reasoning capabilities to generate the future trajectory in textual form, denoted as $T_P$. A post-processing module $\mathcal{P}(\cdot)$ parses the generated text $T_P$ into a numerical trajectory prediction $\mathbf{Y}_{\text{pred}}$, meanwhile, an online feedback mechanism dynamically updates the contextual examples based on the prediction error, forming a closed-loop optimization process.

## 4 MODEL ARCHITECTURE

LLM-TrajEcho designed to enable efficient and real-time trajectory prediction through the seamless integration of visual information and natural language modeling. The framework consists of four main components: RGB video preprocessing and visual feature extraction; natural language transformation of trajectory information and prompt construction; large language model inference and output parsing; and an online feedback mechanism with dynamic context example updates. The overall architecture of the proposed model is illustrated in Figure 1.

### 4.1 RGB VIDEO PREPROCESSING AND VISUAL FEATURE EXTRACTION

In trajectory prediction for autonomous driving, it is essential to capture both local textures and global spatiotemporal dependencies to ensure high accuracy and real-time performance (Ngiam et al., 2022; Shi et al., 2024). To this end, we introduce a patch-wise self-attention encoder. Each video frame is divided into $N$ non-overlapping patches of size $P \times P$, and after learnable projection and positional embedding, the tokens are processed by multi-head self-attention (MHSA) and a feedforward network (FFN). This design enables global receptive fields at low computational cost. Formally, let $\mathbf{X}_t^i \in \mathbb{R}^{P \times P \times 3}$ denote the $i$-th patch of frame $t$, the initial token is constructed as:

$$\mathbf{z}_i^0 = \mathbf{W}_e \text{vec}(\mathbf{X}_t^i) + \mathbf{p}_i, \quad i = 1, \ldots, N, \tag{3}$$

where $\mathbf{W}_e \in \mathbb{R}^{D \times 3P^2}$ is the projection matrix, $\mathbf{p}_i \in \mathbb{R}^D$ is the positional embedding and $\text{vec}(\cdot)$ respresents flattening a patch into a vector. At the $\ell$-th layer, attention is performed as:

$$\mathbf{u}_i^\ell = \sum_{j=1}^{N} \alpha_{ij}^\ell \mathbf{z}_j^\ell, \quad \alpha_{ij}^\ell = \text{softmax}_j\left( \frac{(\mathbf{W}_q \mathbf{z}_i^\ell)^\top (\mathbf{W}_k \mathbf{z}_j^\ell)}{\sqrt{d_k}} \right), \tag{4}$$

and the token is updated via: $\mathbf{z}_i^{\ell+1} = \text{FFN}\left(\text{LN}(\mathbf{z}_i^\ell + \mathbf{u}_i^\ell)\right)$. After stacking $L$ such layers, we apply global average pooling followed by normalization and dimensionality reduction to obtain the frame-level feature:

$$\mathbf{f}_v(t) = \phi\left( \frac{1}{N} \sum_{i=1}^{N} \mathbf{z}_i^L \right). \tag{5}$$

This architecture captures long-range dependencies with a computational complexity of $O(Nd)$, since each patch token attends to a reduced set of $N$ tokens with dimension $d$, rather than to all pixel-level tokens. This patch-wise design avoids quadratic cost in image resolution and thus greatly reduces computation compared to dense attention; this makes it advantageous for online real-time inference (see **Appendix A1**). Finally, a lightweight linear mapping is introduced: $(x_t, y_t) = \mathbf{A}\, \mathbf{f}_v(t) + \mathbf{b}, \quad t = 1, \ldots, T$, where $\mathbf{A}$ and $\mathbf{b}$ are learned by minimizing the regression loss: $\sum_t \|\mathbf{A}\mathbf{f}_v(t) + \mathbf{b} - (x_t, y_t)\|_2^2$. This formulation enables stable extraction of target displacements, providing precise inputs for subsequent prompt construction in natural language.

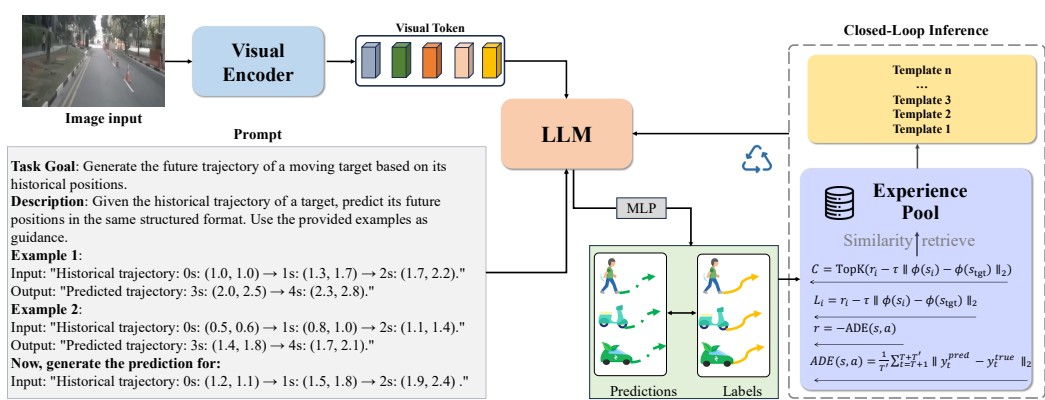

Figure 1: Overview of LLM-TrajEcho. Sequential video frames are encoded by a lightweight vision encoder to extract motion features, which are smoothed and converted into structured language prompts. A LoRA-tuned LLM predicts future trajectories in text, which are parsed into coordinates. An online feedback loop updates the prompt context using recent prediction errors, enabling parameter-free, closed-loop adaptation.

## 4.2 LARGE LANGUAGE MODEL INFERENCE AND OUTPUT PARSING

LLMs possess strong semantic knowledge and contextual reasoning abilities acquired during pretraining. However, full fine-tuning of all parameters incurs substantial computational and storage costs. To address this, we adopt Low-Rank Adaptation (LoRA) within the attention layers, updating only a rank-$r$ increment to the original projection weights $\mathbf{W}_0 \in \mathbb{R}^{d \times d}$:

$$\Delta \mathbf{W} = \mathbf{B}\,\mathbf{A}, \quad \mathbf{A} \in \mathbb{R}^{r \times d},\ \mathbf{B} \in \mathbb{R}^{d \times r},\ r \ll d, \tag{6}$$

the adapted projection becomes $\mathbf{W} = \mathbf{W}_0 + \Delta \mathbf{W}$, which reduces the trainable parameter count from $\mathcal{O}(d^2)$ to $\mathcal{O}(dr)$, while preserving the representational capacity of the pretrained weights.

The constructed prompt is fed into the LoRA-tuned large language model $\mathcal{E}_{LLM}(\cdot)$, which directly generates a natural language description of the future trajectory: $T_P = \mathcal{E}_{LLM}(\text{Prompt})$. The output typically takes the form "Predicted future trajectory: $(\mathrm{x}_{T+1}, \mathrm{y}_{T+1}) \to (\mathrm{x}_{T+2}, \mathrm{y}_{T+2}) \to \cdots$." A post-processing module $\mathcal{P}(\cdot)$ extracts the coordinates from the text using regular expressions or parsers, producing the numerical trajectory: $\mathbf{Y}_{\text{pred}} = \mathcal{P}(T_P)$. From a probabilistic perspective, the output corresponds to the maximum a posteriori estimate of the future motion state:

$$\mathbf{Y}_{\text{pred}} = \arg\max_{\mathbf{Y}} p\big(\mathbf{Y} \mid \text{Prompt}\big), \tag{7}$$

this approach enables the model to retain the contextual reasoning strengths of large-scale LLMs, while supporting efficient online deployment through minimal parameter tuning.

## 4.3 ONLINE FEEDBACK AND DYNAMIC CONTEXT EXAMPLE UPDATE MECHANISM

In dynamic traffic environments, trajectory prediction models need to continuously adapt to evolving conditions (Ngiam et al., 2022). To address this, we design an online feedback mechanism that computes prediction errors in real time and dynamically updates the contextual example set. Let $\mathbf{Y}_{\text{true}}$ denote the ground-truth trajectory and $\mathbf{Y}_{\text{pred}}$ the predicted trajectory. We define the Average Displacement Error (ADE) over $T'$ future time steps as:

$$\text{ADE} = \frac{1}{T'} \sum_{t=T+1}^{T+T'} |\mathbf{y}_t^{\text{pred}} - \mathbf{y}_t^{\text{true}}|_2, \tag{8}$$

a lower ADE indicates higher prediction accuracy. Based on this error, we construct a reward function: $r(s, a) = -\text{ADE}(s, a)$, where lower prediction error corresponds to a higher reward. Each new experience is formulated as a tuple $\{s, a, r(s, a)\}$ and stored in the experience pool $\mathcal{E}$. During

subsequent inference, given the current historical trajectory state $s_{\text{target}}$, the system retrieves relevant examples $\mathcal{E}$ using a similarity-based scoring function. For a candidate example $E = \{s, a, r(s, a)\}$, the relevance score is defined as:

$$L(E, s_{\text{target}}) = r(s, a) - \tau |s - s_{\text{target}}|_2, \tag{9}$$

where $\tau$ is a balance coefficient controlling the trade-off between reward magnitude and state similarity. The top $K$ examples ranked by $L(.)$ are selected to update the context set $C$, ensuring that the prompt remains aligned with current motion patterns.

This online feedback mechanism establishes a closed-loop optimization process, allowing the large language model to maintain strong predictive performance and robustness in dynamically changing environments. Through the theoretical design and mathematical formulation of each module, the LLM-TrajEcho establishes a complete end-to-end modeling pipeline from RGB video to trajectory prediction. It first employs visual encoders and motion parsing techniques to map raw video data into a structured motion representation space. Then, through natural language transformation and prompt construction, it fully activates the contextual learning capabilities of large language models to directly generate future trajectory predictions. Finally, the online feedback mechanism ensures that the model can adapt to dynamic environments, enabling strong performance in real-time, online trajectory forecasting tasks.

## 5 EXPERIMENTS

The main experiments evaluate LLM-TrajEcho on the large-scale benchmark nuScenes Caesar et al. (2020). Beyond trajectory prediction, we further examine the generalization ability of LLM-TrajEcho on camera-based datasets-KITTI Tracking (Geiger et al., 2013), which contain various traffic participants but are not specifically designed for autonomous driving. We provide performence comparison of KITTI Tracking in the main text, additional ablation studies and visualization in the appendix (see **Appendix A4** ). We also investigate how model size affects performance within our proposed framework and analyze the trade-off between performance (see **Appendix A2** ). Meanwhile, we provide results on the Waymo Open Dataset (Sun et al., 2020), a large-scale autonomous driving benchmark (see **Appendix A5** ). These results are intended to further validate the relationship between LLM semantic understanding and numerical prediction, as discussed in our experiments. We consider both nuScenes and Waymo Open to offer comparable data volume and scenario diversity; hence, the Waymo results are included as supplementary material to corroborate our primary findings and hypotheses.

### 5.1 DATASET AND EVALUATION METRICS

nuScenes is one of the most popular autonomous driving dataset, it contains multimodal data and detailed 3D annotations, with 1,000 driving scenes sampled at 2 Hz. The dataset provides a storage approach utilizing JSON metadata alongside separate image files. It spans diverse urban traffic scenarios, including intersections, highways, dense traffic, pedestrians, and adverse weather conditions. KITTI Tracking is a widely used benchmark in object detection and tracking. The dataset contains RGB video sequences captured by onboard cameras, along with annotated target trajectories. The video frame rate is approximately 10 Hz, covering a diverse range of urban roads, highways, and mixed traffic scenes, which introduces rich visual variability and real-world noise.

We evaluate with standard metrics, ADE measures the mean error over all time steps, while FDE evaluates the final position error. Miss Rate (MR) computes the proportion of predictions with endpoint error above a threshold (e.g., 2 m). Results are reported following the official prediction protocol. nuScene provides a 2-second observation window and a 4-second prediction window. For trajectory prediction on the KITTI Tracking dataset, we construct samples from 6-second video clips. Each sample designates the first 3 seconds (30 frames) as the observation window and the subsequent 3 seconds as the prediction horizon. Following the standard split, the dataset is divided into 4,000 training and 1,000 test samples.

Table 1: Comparison on the nuScene dataset Caesar et al. (2020) using the same experimental set up. All baselines are either taken from their official implementations or manually re-implemented following the original papers without ensembling. Bold numbers present the best performance.

| Method | minFDE$_1\downarrow$ | minADE$_5\downarrow$ | minADE$_{10}\downarrow$ | MR$_5$ (%)$\downarrow$ | MR$_{10}$ (%)$\downarrow$ |
|---|---|---|---|---|---|
| THOMAS (Gilles et al., 2021) | 6.71 | 1.33 | 1.04 | 0.55 | 0.42 |
| P2T (Wu et al., 2023) | 10.5 | 1.45 | 1.16 | 0.64 | 0.46 |
| GOHOME (Gilles et al., 2022) | 6.99 | 1.42 | 1.15 | 0.57 | 0.47 |
| LAformer (Liu et al., 2024) | 6.95 | 1.19 | 1.19 | 0.48 | 0.48 |
| MacFormer (Feng et al., 2023) | 7.50 | 1.21 | 0.89 | 0.57 | 0.33 |
| Goal-LBP (Yao et al., 2024b) | 9.21 | 1.02 | 0.93 | 0.32 | 0.27 |
| UniTraj(MTR) (Feng et al., 2024) | 5.41 | 0.96 | 0.84 | 0.43 | 0.41 |
| Demo (Zhang et al., 2024) | 6.59 | 1.22 | 0.89 | 0.43 | 0.34 |
| FiM (Pei et al., 2025) | 6.51 | **0.88** | **0.79** | 0.31 | 0.23 |
| **LLM-TrajEcho (Ours)** | **2.45** | 0.93 | 0.81 | **0.27** | **0.19** |

## 5.2 TRAINING AND INFERENCE SETUP

During training, we fine-tune only the LLM while freezing the pretrained visual encoder for stable feature extraction. To enhance robustness, we apply random cropping, rotation, and color jittering. The model is optimized with AdamW (Loshchilov & Hutter, 2019) (initial learning rate 0.001), and the learning rate is reduced by 0.1 if validation performance does not improve for five epochs. After each epoch, prediction errors are computed via online feedback, and new contextual examples are derived and stored in an experience pool to enable dynamic prompt updates during inference.

During inference, test RGB videos are preprocessed as in training: split into frames and encoded into frame-level features by the visual encoder. These features are parsed into structured trajectories, which are then converted into natural language descriptions. A prompt is formed by combining task instructions with retrieved contextual examples and passed to the fine-tuned LLM for reasoning. The output text is finally parsed into trajectory coordinates by a post-processing module, and performance is evaluated against ground-truth trajectories.

## 5.3 QUANTITATIVE ANALYSIS

On the nuScenes benchmark, LLM-TrajEcho demonstrates compelling advantages in prediction reliability and long-horizon accuracy. As seen in Table 1, it achieves a notably superior minFDE$_1$ of 2.45 m, less than half that of the closest baseline, which strongly validates the effectiveness of the online closed-loop feedback mechanism in curbing endpoint drift. Furthermore, LLM-TrajEcho attains the lowest miss rates across both $K$=5 and 10 settings, underscoring the robustness of its multimodal trajectory generation, a capability we attribute to the LLM's capacity for global semantic reasoning. While top-performing methods such as FiM (Pei et al., 2025) excel in short-term displacement error (minADE), our approach remains highly competitive (minADE$_5$: 0.93, minADE$_{10}$: 0.81), illustrating that a vision-based encoding coupled with LLM-based inference can capture spatiotemporal dynamics effectively without relying on additional sensor modalities.

We evaluate LLM-TrajEcho on KITTI Tracking under $K = 5$ settings against representative baselines (Table 2). Our method achieves the best minFDE$_5$ of 2.46 m and the lowest MR$_5$ of 0.37, while maintaining a strong minFDE$_5$ of 1.24. This specific performance profile offers direct validation of our core design choices. The superior minFDE and significantly low MR directly demonstrate the efficacy of our online closed-loop feedback in mitigating endpoint error accumulation, while the strong minADE confirms that our vision backbone with LLM reasoning effectively captures spatiotemporal dynamics. Furthermore, The balanced performance further highlights the LLM paradigm's advantage over specialized approaches—avoiding feature redundancy in raster-based methods (P2T (Wu et al., 2023), GOHOME (Gilles et al., 2022))and overcoming limited global semantics in interaction-aware models (AgentFormer (Yuan et al., 2021), MHA-JAM (Messaoud et al., 2021)).

Finally, benchmark results confirming significantly faster inference than conventional transformers underscore the practical efficiency gained from our lightweight design choices, solidifying LLM-TrajEcho's suitability for real-time deployment. It is important to note that our inference perfor-

Table 2: Performance comparison under $K = 5$ settings on the KITTI Tracking datests (Geiger et al., 2013) using the same experimental set up. All baselines are either taken from their official implementations or manually re-implemented following the original papers without ensembling. Bold numbers present the best performance.

| Method | minADE$_5$ (m) ↓ | minFDE$_5$ (m) ↓ | MR$_5$ (%) ↓ |
|---|---|---|---|
| Trajectron++ (Salzmann et al., 2020) | 1.88 | 4.00 | 0.70 |
| GATraj (Cheng et al., 2023) | 1.87 | 4.08 | 0.65 |
| SG-Net (Liu et al., 2021a) | 1.85 | 3.87 | 0.68 |
| MHA-JAM (Messaoud et al., 2021) | 1.81 | 3.72 | 0.59 |
| AgentFormer (Yuan et al., 2021) | 1.59 | 3.14 | 0.62 |
| P2T (Wu et al., 2023) | 1.45 | 3.80 | 0.64 |
| GOHOME (Gilles et al., 2022) | 1.42 | 3.70 | 0.57 |
| CASPNet (Xiong et al., 2023) | 1.41 | 3.60 | 0.60 |
| MUSE-VAE (Lee et al., 2022) | 1.38 | 2.90 | 0.58 |
| THOMAS (Gilles et al., 2021) | 1.33 | 2.92 | 0.55 |
| HLSTrajForecast (Choi & Min, 2022) | 1.33 | 2.92 | 0.60 |
| PGP (Deo et al., 2022) | 1.27 | 2.70 | 0.52 |
| LAformer (Liu et al., 2024) | 1.19 | 2.50 | 0.48 |
| FRM (Distelzweig et al., 2024) | **1.18** | 2.48 | 0.48 |
| **LLM-TrajEcho (Ours)** | 1.24 | **2.46** | **0.37** |

Table 3: Inference time comparison of nuScene Caesar et al. (2020) on NVIDIA A100. The results were recorded for each sample, and the best four comparisons were selected.

| Methods | Batch size = 1 ↓ | | | Batch size = 4 ↓ | | | Batch size = 8 ↓ | | |
|---|---|---|---|---|---|---|---|---|---|
| | FPS | Avg. Inference (ms) | Per sample (ms) | FPS | Avg. Inference (ms) | Per sample (ms) | FPS | Avg. Inference (ms) | Per sample (ms) |
| UniTraj(MTR) (Feng et al., 2024) | 67.8 | 14.75 ± 1.84 | 14.75 | 258.3 | 15.48 ± 2.84 | 3.87 | 512.6 | 15.60 ± 1.87 | 1.95 |
| GOHOME (Gilles et al., 2022) | 85.4 | 11.71 ± 1.46 | 11.71 | 325.8 | 12.27 ± 2.25 | 3.07 | 654.2 | 12.23 ± 1.47 | 1.53 |
| P2T (Wu et al., 2023) | 103.2 | 9.69 ± 1.21 | 9.69 | 395.6 | 10.11 ± 1.85 | 2.53 | 812.4 | 9.85 ± 1.32 | 1.23 |
| **LLM-TrajEcho (Ours)** | **231.7** | **4.32 ± 0.53** | **4.32** | **923.0** | **4.33 ± 0.93** | **1.08** | **1887.5** | **4.24 ± 0.51** | **0.53** |

mance of 231.7 FPS was measured on an NVIDIA A100 GPU under single-frame inference conditions. While this represents the theoretical upper bound of our framework's performance, practical deployment scenarios typically involve continuous video stream processing on edge devices. To better reflect real-world applicability, we additionally evaluated our model on an NVIDIA RTX 4090, which more closely matches the computational capabilities of modern high-performance edge computing platforms (see **Appendix A3** ). In summary, the results robustly validate that our integrated architecture of vision-based encoding, LLM-based reasoning, and online correction collectively enables a new state of the art in efficient and robust trajectory prediction.

## 5.4 CASE STUDY

Visualizations on the nuScenes dataset demonstrate LLM-TrajEcho's superior performance in trajectory smoothness and accuracy. A key finding is that the model autonomously learns to correlate numerical predictions with semantic driving concepts through the closed-loop process. As illustrated in Figure 2, when the motion pattern predictor is active (yellow box, top-left), LLM-TrajEcho accurately forecasts turning angles at intersections, closely aligning with the ground truth. This capability stems from the model's ability to interpret the structured language descriptions of visual features. The analysis of prediction logs reveals a consistent correlation between the sign of positional coordinates (positive/negative values) and the semantically reasoned turning decisions (e.g., "Left" or "Right"). This indicates that the LLM has developed an implicit understanding of vehicle kinematics and its corresponding textual representation without explicit human supervision.

The critical role of visual input is further substantiated by an ablation study (see **Appendix A3** ). . When visual features are replaced with raw numerical trajectories and sensor data, the LLM fails to develop a grounded internal representation of the historical scene context. Consequently, it regresses to generating only plausible text without accurately predicting the vehicle's true future motion. This contrast confirms that the spatiotemporal semantics embedded in the visual features are essential for enabling the LLM to function as a capable trajectory predictor, rather than a mere language model.

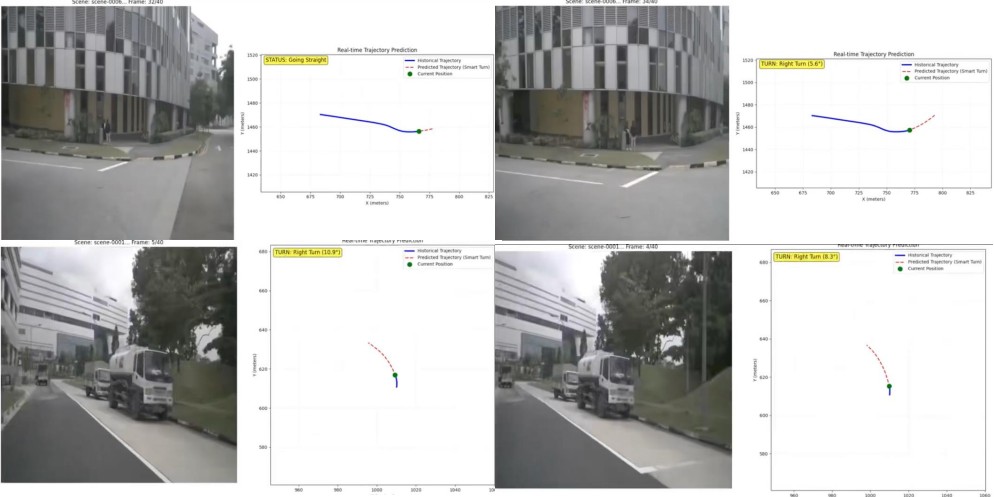

Figure 2: Visualizations of LLM-TrajEcho on nuScene dataset Caesar et al. (2020). The yellow box indicates the predicted motion status: turning angle and direction.

Table 4: Impact of Each Module on LLM-Traj Performance (nuScene Caesar et al. (2020))

| Configuration | minFDE$_1 \downarrow$ | minADE$_5$ (m)$\downarrow$ | minADE$_{10}$ (m) $\downarrow$ | MR$_5$ (%) $\downarrow$ | MR$_{10}$ (%)$\downarrow$ |
|---|---|---|---|---|---|
| **Full Model** | **2.45** | **0.93** | **0.81** | **0.27** | **0.19** |
| w/o NLP | 2.56 | 0.99 | 0.85 | 0.32 | 0.22 |
| w/o Online Feedback | 2.61 | 0.99 | 0.87 | 0.34 | 0.23 |
| w/o Example Update | 2.59 | 0.97 | 0.88 | 0.33 | 0.24 |

## 5.5 ABLATION STUDY

The analysis based on ablation experiment results shows that all components in LLM traj echo have important contributions to performance improvement. As demonstrated in Table 4, the removal of the NLP module led to a significant decline in various indicators, especially the increase of minADE$_5$ from 0.93 to 0.99, which confirmed the key role of natural language description in transforming visual motion features into LLM-intelligible semantics, which provided a structured scene reasoning basis for the model. The lack of online feedback mechanism made minFDE$_1$ rise to 2.61, indicating that the module effectively suppressed the cumulative deviation in trajectory prediction through real-time error correction, and improved the endpoint accuracy and scene adaptability. After canceling the example update module, the long-term prediction index minADE$_{10}$ deteriorated to 0.88, reflecting its ability to enhance the model's understanding of complex interaction history through dynamic experience reuse. The three work together to support the accuracy and robustness of the framework from semantic mapping, error constraint and context learning.

## 6 CONCLUSION

LLM-TrajEcho presents a lightweight, fully closed-loop framework for trajectory prediction by synergistically integrating visual encoding, LLM-based reasoning, and online feedback. Our work demonstrates that pure-vision inputs combined with efficient LLM inference can achieve accuracy comparable to fusion-based methods while enabling real-time performance. The proposed online prompt-update mechanism continuously refines predictions without costly retraining, offering a practical path toward deployable autonomous systems. Experiments confirm significant improvements in prediction accuracy, robustness, and inference efficiency, establishing a new paradigm for adaptive trajectory forecasting. Furthermore, the architecture's reliance on visual input and its flexible prompt-based interface suggest strong potential for generalization to robotic vision tasks, such as navigation and manipulation, where similar requirements for real-time adaptation and scene understanding exist.

## 7 ETHICS STATEMENT

This work does not raise any ethical concerns. It does not involve human or animal subjects, personal or sensitive data, or experiments requiring IRB approval. No datasets with privacy issues, biases, or harmful content were used. The methods and applications described in this paper do not present foreseeable risks of misuse or negative societal impact. The authors have adhered to the ICLR Code of Ethics throughout the research and submission process.

## 8 REPRODUCIBILITY STATEMENT

We have made extensive efforts to ensure the reproducibility of our work. All model architectures, training procedures, and hyperparameters are described in detail in the main text. Additional implementation details and ablation studies are provided in the appendix. The datasets used in our experiments are publicly available, and we document the preprocessing steps to facilitate replication. Pseudocode and theoretical derivations are included where relevant to clarify our methods. Furthermore, we provide an anonymous link to the source code and scripts in the supplementary materials to enable others to reproduce our results.

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

# A  APPENDIX

This appendix provides supplementary materials that complement the main text. It includes detailed derivations of the mathematical formulations presented in the method section, as well as additional experimental results and visualizations in experiment section. The derivations clarify the intermediate steps omitted in the main paper for brevity, while the extra experiments further validate the effectiveness and robustness of our proposed approach.

**The Use of Large Language Models (LLMs).** This paper acknowledges the use of a large language model (LLM) during its writing process. The LLM did not contribute to the scientific novelty, methodological design, empirical evaluations, or any other scholarly aspects of this work. The authors have meticulously reviewed and edited all LLM-generated suggestions and retain full authorship and accountability for the work presented herein.

## A.1  COMPLEXITY ANALYSIS

Consider an $H \times W$ RGB frame with $M = HW$ pixels. Pixel-level self-attention requires $O(M^2 d) = O(H^2 W^2 d)$ operations. By dividing the frame into $N = HW/P^2$ non-overlapping $P \times P$ patches, we obtain $N$ tokens of dimension $d$. Embedding costs $O(Nd)$, and naive MHSA over $N$ tokens would be $O(N^2 d)$. To reduce the quadratic term, we adopt $O(N)$ attention mixing (patch-wise tokenization), where each token attends to only a constant set of neighbors. This yields

$$\mathcal{C}_{\text{attn}} = O(Nd) = O\left( \tfrac{HW}{P^2}\, d \right),$$

linear in both token count and head dimension. This patch-wise design therefore avoids quadratic growth in image resolution, enabling real-time inference, while still modeling long-range dependencies efficiently. **Back to main text**

## A.2  PERFORMANCE ANALYSIS OF DIFFERENT MODEL SIZE

Our experiments reveal a noteworthy finding regarding model scalability in trajectory prediction. As illustrated in Table 5, the performance gap between compact models (e.g., Qwen2.5-0.5B, GPT-2 Medium) and larger, more advanced models (e.g., Qwen3-7B) is marginal on metrics such as minADE and minFDE. This suggests that the numerical regression nature of trajectory prediction tasks exhibits a form of performance saturation with respect to model size; beyond a certain threshold, additional parameters yield diminishing returns on accuracy.

The critical differentiator, therefore, shifts from pure predictive performance to computational pragmatism. Smaller LLMs, when coupled with an efficient visual encoder, achieve an optimal balance between accuracy and latency, making them ideally suited for on-edge deployment where resources are constrained. In contrast, larger LLMs, with their superior semantic reasoning capabilities, are better deployed in a cloud-based setting to handle higher-level tasks such as strategic planning and explainable decision-making. This delineation provides a clear guideline for practical system design: use compact, specialized models for real-time perception/prediction loops, and reserve larger, generalist models for asynchronous, high-level reasoning. **Back to main text**

Table 5: Comparison model side on the nuScene dataset (Caesar et al., 2020) using the same experimental set up.

| Method | minFDE$_1\downarrow$ | minADE$_5\downarrow$ | minADE$_{10}\downarrow$ | MR$_5$ (%)$\downarrow$ | MR$_{10}$ (%)$\downarrow$ |
|---|---|---|---|---|---|
| GPT-2 medium 335M | 2.88 | 1.11 | 0.89 | 0.27 | 0.26 |
| llama 3.2 1B | 2.64 | 1.02 | 0.83 | 0.31 | 0.25 |
| Qwen 2.5 0.5B | 2.41 | 0.96 | 0.84 | 0.29 | 0.23 |
| Qwen 2.5 3B | 2.49 | 1.02 | 0.81 | 0.31 | 0.24 |
| llama 3.2 3B | 2.59 | 1.08 | 0.82 | 0.27 | 0.23 |
| **Qwen3 7B** | **2.45** | **0.93** | **0.81** | **0.27** | **0.19** |

Table 6: Inference time comparison: Simulated Real-world (RTX 4090) on nuScene dataset (Caesar et al., 2020)

| Methods | Batch size = 1 $\downarrow$ | | | Batch size = 4 $\downarrow$ | | | Batch size = 8 $\downarrow$ | | |
|---|---|---|---|---|---|---|---|---|---|
| | FPS | Avg. Inference (ms) | Per sample (ms) | FPS | Avg. Inference (ms) | Per sample (ms) | FPS | Avg. Inference (ms) | Per sample (ms) |
| P2T (Wu et al., 2023) | 11.3 | $88.5 \pm 11.1$ | 88.5 | 13.4 | $74.6 \pm 13.7$ | 18.7 | 18.1 | $55.2 \pm 7.4$ | 6.9 |
| GOHOME (Gilles et al., 2022) | 9.3 | $107.5 \pm 13.4$ | 107.5 | 11.0 | $90.9 \pm 16.7$ | 22.7 | 14.6 | $68.5 \pm 8.2$ | 8.6 |
| UniTraj(MTR) (Feng et al., 2024) | 7.4 | $135.1 \pm 16.9$ | 135.1 | 8.7 | $114.9 \pm 21.1$ | 28.7 | 11.4 | $87.7 \pm 10.5$ | 11.0 |
| **Ours (RTX 4090 - Simulated)** | **32.3** | **$39.5 \pm 4.9$** | **39.5** | **35.2** | **$32.1 \pm 4.7$** | **8.0** | **42.0** | **$23.8 \pm 2.9$** | **3.0** |

## A.3 MORE EVALUATION ON NUSCENE

**Inference time.** We primarily report the performance under a batch size of 1, as it most accurately reflects the real-time inference capability required for onboard deployment where data arrives sequentially. Results with larger batch sizes are also provided to demonstrate the model's scalability in offline processing scenarios.

Based on the inference time comparison conducted on the NVIDIA RTX 4090, LLM-TrajEcho demonstrates significant efficiency advantages under video stream input settings. As illstrate in Table 6, when using a batch size of 1, our method achieves 32.3 FPS, which is 2.9× faster than the strongest baseline, P2T (11.3 FPS), with a per-sample latency of only 39.5 ms, well satisfying the real-time requirements of autonomous driving. As the batch size increases to 8, LLM-TrajEcho's throughput rises to 42.0 FPS while maintaining a per-sample processing time of 3.0 ms, highlighting the framework's excellent scalability. This performance gain stems from our carefully designed lightweight architecture: the patch-wise self-attention encoder reduces computational complexity from quadratic to linear, and the LoRA fine-tuning strategy significantly cuts the number of trainable parameters. In contrast to rasterization-based methods (e.g., P2T (Wu et al., 2023), GOHOME (Gilles et al., 2022)) and interaction-aware models (e.g., UniTraj (Feng et al., 2024)), our LLM-driven framework maintains high prediction accuracy while achieving more efficient computational resource utilization by avoiding redundant feature computations and optimizing memory access patterns, thereby offering a reliable foundation for practical edge deployment.

**More Ablation.** To further validate our hypothesis on the role of visual semantics, we conducted an ablation study where only ego status (position, velocity, and heading) was provided as numerical input to the LLM, without any visual data. As shown in Table 7, the "ego status only" configuration significantly underperforms across all metrics compared to the full visual-LMM model and other baselines. This result underscores a fundamental limitation of LLMs in trajectory prediction: as probabilistic models primarily trained on textual correlations, they struggle to infer implicit spatiotemporal representations from pure numerical sequences. Without grounded visual context, the LLM fails to capture the environmental dynamics necessary for accurate trajectory forecasting, effectively reducing its output to plausible but physically ungrounded text.

The performance gap highlights the critical role of visual feature extraction, which acts as "eyes" for the LLM by converting raw pixels into structured semantic descriptions. By integrating image-derived features, our framework enables the model to interpret scene layout, interaction patterns, and motion constraints—elements indispensable for reasoning about future trajectories. These findings confirm that while LLMs excel at linguistic reasoning, their application to numerical regression tasks like trajectory prediction requires complementary visual grounding to bridge the gap between token probabilities and physical world dynamics. **Back to main text**

Table 7: Comparison on the nuScene dataset Caesar et al. (2020) using ego status input only. All baselines are either taken from their official implementations or manually re-implemented following the original papers without ensembling. Bold numbers present the best performance.

| Method | minFDE$_1\downarrow$ | minADE$_5\downarrow$ | minADE$_{10}\downarrow$ | MR$_5$ (%)$\downarrow$ | MR$_{10}$ (%)$\downarrow$ |
|---|---|---|---|---|---|
| THOMAS (Gilles et al., 2021) | 6.71 | 1.33 | 1.04 | 0.55 | 0.42 |
| P2T (Wu et al., 2023) | 10.5 | 1.45 | 1.16 | 0.64 | 0.46 |
| GOHOME (Gilles et al., 2022) | 6.99 | 1.42 | 1.15 | 0.57 | 0.47 |
| LAformer (Liu et al., 2024) | 6.95 | 1.19 | 1.19 | 0.48 | 0.48 |
| MacFormer (Feng et al., 2023) | 7.50 | 1.21 | 0.89 | 0.57 | 0.33 |
| Goal-LBP (Yao et al., 2024b) | 9.21 | 1.02 | 0.93 | 0.32 | 0.27 |
| UniTraj(MTR) (Feng et al., 2024) | 5.41 | 0.96 | 0.84 | 0.43 | 0.41 |
| Demo (Zhang et al., 2024) | 6.59 | 1.22 | 0.89 | 0.43 | 0.34 |
| FiM (Pei et al., 2025) | 6.51 | **0.88** | **0.79** | 0.31 | 0.23 |
| **LLM-TrajEcho (ego status only)** | 9.82 | 2.01 | 1.67 | 0.58 | 0.49 |
| **LLM-TrajEcho (Vision-based)** | **2.45** | 0.93 | 0.81 | **0.27** | **0.19** |

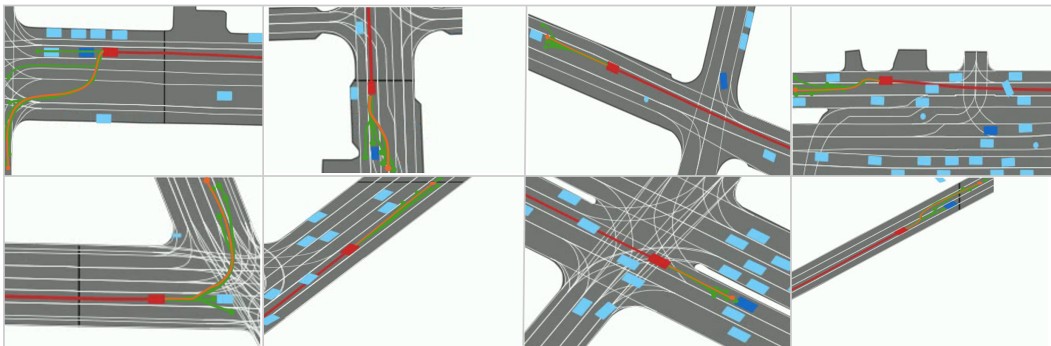

Figure 3: Quantitative results on the KITTI tracking dataset Geiger et al. (2013). The red lines represent observed (historical) trajectories, orange lines indicate the ground-truth future trajectories, and green lines denote the predicted results by LLM-TrajEcho.

## A.4 MORE EVALUATION ON KITTI TRACKING

**Ablation Study.** To quantitatively assess the contribution of each module to the overall performance of LLM-TrajEcho, we conduct an ablation study on the KITTI Tracking dataset. We evaluate the effects of removing (1) the natural language conversion module (w/o NLP), (2) the online feedback mechanism (w/o FB), and (3) the dynamic example update module (w/o Update). Performance is measured in terms of minADE$_6$, minFDE$_6$, MR$_6$. As shown in Table 8, the complete model achieves 1.20 m / 1.75 m / 15.2% on these three metrics. Removing the natural language conversion results in a significant performance drop, with ADE increasing to 1.35 m (+12.5%), FDE to 1.90 m (+8.6%), and MR to 17.5% (+15.1%), indicating the critical role of this module in activating the contextual reasoning capabilities of the LLM. Without the online feedback mechanism, ADE rises to 1.30 m (+8.3%), FDE to 1.82 m (+4.0%), and MR to 16.8% (+10.5%), demonstrating that real-time error feedback effectively mitigates prediction drift. Disabling dynamic example updates yields ADE of 1.32 m, FDE of 1.85 m, and MR of 17.0%, reflecting respective increases of 10.0%, 5.7%, and 11.8% over the full model. These results confirm that example updates improve the model's adaptability to evolving motion patterns. Overall, the ablation study validates the complementary effect of the three components in enhancing accuracy, stability, and generalization of the proposed framework.

**Qualitative Results Analysis.** To provide an intuitive understanding of the model's behavior in complex traffic environments, we conduct qualitative analysis on several representative scenarios. Figure 3 presents a visual comparison between the predicted candidate trajectories generated by LLM-Traj and the corresponding ground-truth trajectories in scenes involving sharp turns, multi-agent interactions, and complex intersections. The results show that LLM-Traj not only closely follows the ground truth in terms of global trajectory direction, but also produces smooth and contin-

Table 8: Ablation Study: Impact of Each Module on LLM-Traj Performance (KITTI Tracking)

| Configuration | minADE$_5$ (m)↑ | minFDE$_5$ (m) ↑ | MR$_5$ (%) ↑ |
|---|---|---|---|
| **Full Model** | **1.20** | **1.75** | **15.2** |
| w/o NLP | 1.35 | 1.90 | 17.5 |
| w/o Online Feedback | 1.30 | 1.82 | 16.8 |
| w/o Example Update | 1.32 | 1.85 | 17.0 |

Table 9: Comparison on the Waymo open dataset Sun et al. (2020). P&M represent using ego status input only, E2E include vison input. All baselines are either taken from their official implementations or manually re-implemented following the original papers without ensembling. Bold numbers present the best performance.

| Method | minFDE↓ | minADE↓ | miss Rate↓ | soft mAP↑ |
|---|---|---|---|---|
| HDGT (Jia et al., 2023b) | 0.7676 | 1.1077 | 0.1325 | 0.3709 |
| MPA (Konev, 2022) | 0.5913 | 1.2507 | 0.1603 | 0.3930 |
| MTR (Shi et al., 2022) | 0.6050 | 1.2207 | 0.1351 | 0.4216 |
| Wayformer factorized (Nayakanti et al., 2022) | **0.5447** | 1.1255 | 0.1229 | 0.4260 |
| Wayformer multi-axis (Nayakanti et al., 2022) | 0.5454 | 1.1280 | 0.1228 | 0.4335 |
| MTR-A (Shi et al., 2024) | 0.5640 | 1.1344 | 0.1160 | 0.4594 |
| MotionLM (Seff et al., 2023) | 0.5509 | **1.1199** | **0.1058** | 0.4507 |
| **LLM-TrajEcho (P&M)** | 1.7981 | 1.2535 | 0.1850 | 0.3120 |
| **LLM-TrajEcho (E2E)** | 0.5785 | 1.1928 | 0.1180 | **0.4618** |

uous curves at the local level. In scenarios with high uncertainty, the model is capable of generating multiple plausible trajectory candidates and leveraging the online feedback mechanism to automatically select the most reliable one. This behavior reflects the strength of large language models in modeling long-range dependencies and capturing multimodal dynamics. **Back to main text**

## A.5 EVALUATION ON WAYMO OPEN

**Ablation Study.** We further validated our approach on the Waymo Open Dataset (Sun et al., 2020) to demonstrate its generalization capability. Specifically, we conducted comparative experiments using two distinct subsets: the Perception and Motion (P&M) Dataset, which provides high-resolution raw sensor data with detailed 3D bounding box annotations and map information, and its derivative End-to-End (E2E) Dataset containing video inputs for driving policy learning. This experimental design allows direct comparison between numerically driven inputs (from P&M) and vision-based inputs (from E2E) within identical environmental contexts.

As shown in Table 9, the results consistently reinforce our findings from nuScenes: models relying solely on numerical trajectory data from P&M significantly underperform compared to those utilizing visual features from E2E. This performance gap persists despite P&M's rich annotation scheme, confirming that even detailed numerical representations cannot compensate for the absence of raw visual perception. The LLM's inherent limitation as a probabilistic text model becomes apparent when deprived of visual grounding—it fails to develop a meaningful understanding of spatial relationships and dynamic interactions from numerical abstractions alone.

These cross-dataset experiments substantiate that visual feature extraction provides indispensable semantic context for trajectory prediction, transcending what can be achieved through numerical data alone. The consistent outcomes across both nuScenes and Waymo benchmarks strengthen the generalizability of our conclusion that LLMs require visual "eyes" to effectively bridge the gap between textual reasoning and physical trajectory forecasting. **Back to main text**

**Inference Time.** Our inference time evaluation on the Waymo Open Dataset (Table 10) demonstrates that LLM-TrajEcho achieves significant computational efficiency compared to state-of-the-art trajectory prediction methods. With an average inference latency of 27.8 ms and a throughput of 36.93 FPS, our framework substantially outperforms all baselines, meeting the real-time requirements (>30 FPS) for autonomous driving systems.

Table 10: Inference analysis on the Waymo open dataset Sun et al. (2020). All baselines are either taken from their official implementations or manually re-implemented following the original papers without ensembling. Bold numbers present the best performance.

| Method | Avg.inference (ms)↓ | Max inference (ms)↓ | Min inference (ms)↓ | FPS↓ |
|---|---|---|---|---|
| HDGT (Jia et al., 2023b) | $68.5 \pm 8.2$ | 89.1 | 52.3 | 14.6 |
| MPA (Konev, 2022) | - | - | - | - |
| MTR (Shi et al., 2022) | $45.2 \pm 5.7$ | 58.9 | 36.1 | 22.1 |
| Wayformer factorized (Nayakanti et al., 2022) | $38.7 \pm 4.9$ | 49.3 | 30.5 | 25.8 |
| Wayformer multi-axis (Nayakanti et al., 2022) | $51.4 \pm 6.3$ | 66.8 | 40.2 | 19.5 |
| MTR-A (Shi et al., 2024) | $41.8 \pm 5.2$ | 54.3 | 33.1 | 23.9 |
| MotionLM (Seff et al., 2023) | $88.9 \pm 11.3$ | 115.6 | 68.4 | 11.2 |
| **LLM-TrajEcho (Ours)** | $\mathbf{27.8 \pm 4.33}$ | **38.99** | 20.29 | **36.93** |

The efficiency gains are primarily attributed to our lightweight architectural design: the patch-wise self-attention encoder reduces spatial-temporal modeling complexity from quadratic to linear, while LoRA-based fine-tuning minimizes trainable parameters without compromising representational capacity. In contrast, graph-based approaches (e.g., HDGT (Jia et al., 2023b), 68.5 ms) and multi-path architectures (e.g., MTR (Shi et al., 2022), 45.2 ms) incur high computational overhead due to their structural complexity. Notably, language-model-based methods like MotionLM (Seff et al., 2023) (88.9 ms) suffer from sequential decoding latency, whereas our method avoids this bottleneck by integrating visual features with LLM reasoning in a parallelizable manner.

These results underscore the practical viability of LLM-TrajEcho for edge deployment, balancing predictive accuracy with stringent latency constraints. The consistent efficiency advantage across metrics highlights the scalability of our approach in real-world applications.

## A.6 MORE VISUALZATION

We provide extensive qualitative results showcasing LLM-TrajEcho's performance across diverse driving scenarios in both nuScenes and Waymo Open datasets. The visualizations demonstrate our model's capability to generate accurate and socially-compliant trajectories in complex urban environments, including intersections, highway merges, and pedestrian-heavy scenarios.

On nuScenes, LLM-TrajEcho consistently produces smooth trajectories that adhere to road geometry . The model shows particular strength in long-horizon predictions (up to 4 seconds), where the online feedback mechanism effectively corrects accumulating errors through dynamic prompt updates.

Waymo evaluations further reveal the framework's adaptability to different sensor configurations and annotation protocols. The visual comparisons highlight how our method maintains stable performance under worse lighting conditions and traffic densities, with the structured language representations providing consistent reasoning patterns across datasets.

These qualitative results, available in the supplementary material, complement our quantitative findings and offer valuable insights into the model's decision-making process. The visual evidence underscores LLM-TrajEcho's practical utility for real-world deployment where interpretability and robustness are paramount.

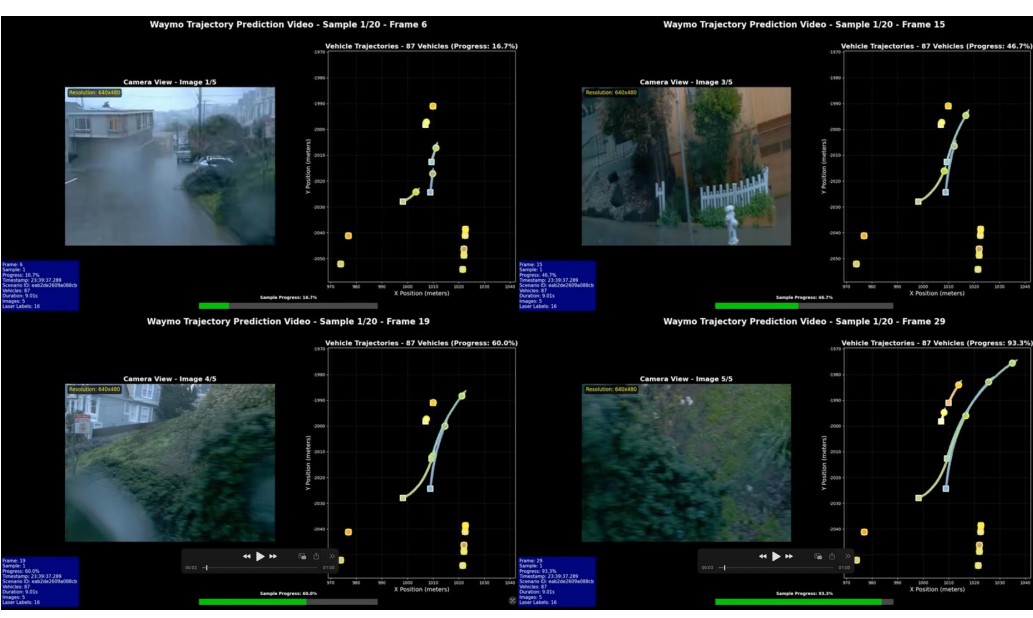

Figure 4: More quantitative results on the Waymo open dataset (Sun et al., 2020).

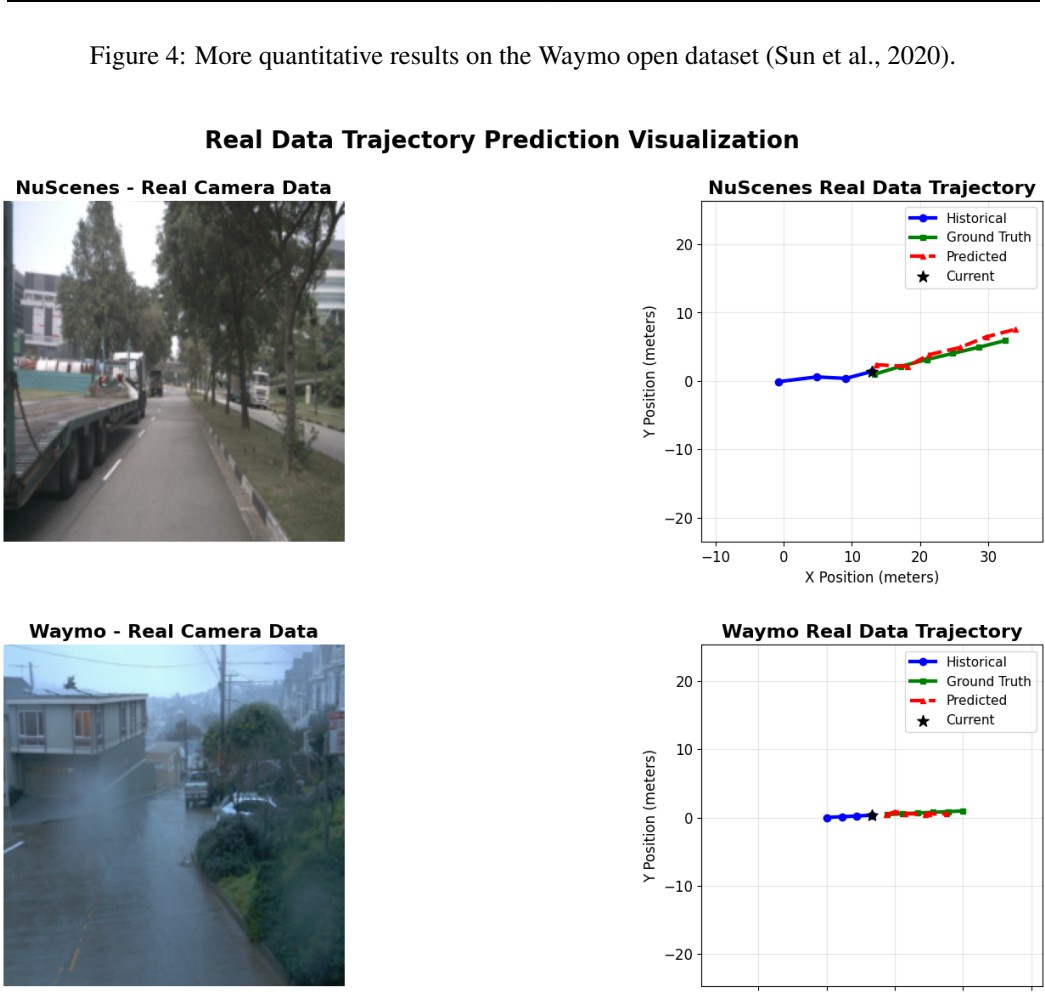

Figure 5: More quantitative results on the nuScene dataset Caesar et al. (2020).

