# OpenReview forum: "Prompt, Predict, Correct: LLM-TrajEcho for Closed-Loop Trajectory Forecasting via Online Prompt Feedback"
_ICLR.cc/2026/Conference — ICLR 2026 Conference Withdrawn Submission_

### Official Review · Reviewer_gErW · 2025-10-30

**Soundness:** 2
**Presentation:** 3
**Contribution:** 3
**Rating:** 4
**Confidence:** 4

**Summary:**

This paper proposes LLM-TrajEcho, a lightweight vision–language-integrated closed-loop trajectory forecasting framework for autonomous driving. The method converts visual motion cues from video into structured natural language prompts, which are then processed by a LoRA-tuned large language model (LLM) to predict future trajectories. A novel online feedback mechanism updates contextual examples dynamically based on prediction errors, allowing real-time, gradient-free model adaptation. Experiments on nuScenes, KITTI, and Waymo Open datasets demonstrate strong improvements in final displacement error (FDE), miss rate, and inference speed, suggesting a promising path toward interpretable and efficient LLM-driven motion forecasting

**Strengths:**

The idea of using highly compressed semantic information instaed of redundancy and costly sensor data is good. And the online feedback mechanism, gives the agent self-envloving ability, is interesting.
The proposed method can predicted better trajectories than other method, and get higher minFDE and miss rate.

**Weaknesses:**

The paper’s closed-loop feedback mechanism is conceptually interesting but not thoroughly validated — experiments on open-loop benchmarks do not convincingly demonstrate real-time adaptation.
The methodological foundations behind the feedback and reward design are insufficiently rigorous, relying on heuristic formulations without clear theoretical support.
The experimental transparency is limited — key details about the LLM model, training setup, and baseline reproduction are missing, making it difficult to assess reproducibility and verify claimed efficiency.

**Questions:**

1. Line 66 - 69. The third challenge, "Current LLM-based methods generally remain static during inference", why static? "Approaches such as autoregressive decoding or distillation still require offline gradient updates", why do they require offline gradient updates?
2. About the close-loop. In this paper, the author emphasis close-loop, but in the experiment, the author only provide results on open-loop benchmark like nuScenes and KITTI.
3. About the reward function. You define the reward function as $r(s,a)=-ADE(s, a)$, but it seems that ADE only used a. Where is the state? And you define the state s as "current historical trajectory", I think this state is so compressd that some important information like the perceptioned information are filtered. Just imagine that a car is driving smoothly (your history trajectory is keeping forward), but a man appears in front of the car, without the perception information, the car using your algorithm to conduct motion prediction will collide with the man, which is dangerous.
4. It seems that you actually only generate the trajectories of the ego car based on the input video, and then pass the prompt with trajectory to LLM, right? What confused me is, why don't you use history waypoint provided by the dataset itself? As far as I know, nuscenes provide these data items. In that way, you do not even need video input.
5. About the experiment results in Table 3. In the paper "UniTraj", the author seems to not provide the inference time. Do you reproduce the algorithm on your server? Let's see the column batch_size=1. I suspect the result, as your method (using LLM) gets a higher FPS than traditional methods. And I think you should provide informations of the LLM you utilize in this paper, like:
  1) what LLM do you use?
  2) the parameter size (3B or 6B)?
  3) do you use the official ckpts? Do you finetuning it? Is so, using what dataset?


Typos:
1. Line 154, the last $D_v$ of the line.

---

### Official Review · Reviewer_pw4V · 2025-10-30

**Soundness:** 3
**Presentation:** 3
**Contribution:** 3
**Rating:** 4
**Confidence:** 3

**Summary:**

The paper presents LLM-TrajEcho, a novel end-to-end, vision-based framework for trajectory prediction in autonomous driving. The core idea is to leverage a Large Language Model (LLM) to directly predict future trajectories by translating spatiotemporal features from video sequences into structured natural language prompts. A key innovation is the introduction of an online, gradient-free, closed-loop feedback mechanism. This mechanism dynamically refines the LLM's input prompt by retrieving relevant examples from an "experience pool" based on past prediction errors and state similarity, allowing the model to adapt to changing conditions and mitigate error accumulation without requiring online parameter updates. The framework employs a lightweight visual encoder for efficiency and uses Parameter-Efficient Fine-Tuning (LoRA) to adapt the LLM. Experiments on the nuScenes and KITTI datasets show that the proposed method achieves competitive performance, particularly in long-term prediction metrics, while maintaining very high inference speeds suitable for real-time deployment.

**Strengths:**

*   The paper is well-written, clearly structured, and easy to understand. The motivation is well-established, and the proposed architecture and mechanisms are described in sufficient detail to grasp the core concepts.
*   The primary contribution, an online gradient-free feedback mechanism that leverages an LLM's in-context learning for real-time adaptation, is novel and significant. It presents a clever, parameter-free approach to address the challenge of error accumulation in dynamic environments.
*   The framework is designed with a strong emphasis on computational efficiency, utilizing a lightweight patch-wise attention encoder and LoRA. The reported inference speeds validate its potential for real-world, on-device deployment.
*   The empirical results are strong, demonstrating competitive performance against state-of-the-art methods. The significant improvements in long-horizon metrics like FDE and MR provide direct evidence for the effectiveness of the proposed closed-loop correction mechanism.
*   The ablation studies are well-conducted and effectively demonstrate the necessity of each key component of the proposed framework (the NLP module, the online feedback, and the example update mechanism), thereby strengthening the paper's claims.

**Weaknesses:**

*   The practical implementation and scalability of the "experience pool" are not fully addressed. The paper does not specify how the pool is managed over long-term operation, which could lead to unbounded growth and a potential bottleneck in the retrieval process.
*   While the paper mentions a "lightweight patch-wise self-attention encoder," specific architectural details (e.g., number of layers, heads, total parameters) are not provided, which could make exact replication of the visual frontend challenging.
*   The mechanism for parsing the LLM's textual output back into numerical coordinates relies on regular expressions, which can be brittle and prone to failure if the LLM's output format deviates even slightly.

**Questions:**

With the questions below properly addressed, I would be inclined to maintain or raise my rating.
1.  Could the authors elaborate on the long-term management strategy for the experience pool? Is its size fixed, and if so, what is the replacement policy (e.g., FIFO, reward-based)? If its size is not fixed, how is the potential increase in retrieval latency addressed to maintain real-time performance?
2.  Could the authors provide more specific architectural details for the "patch-wise self-attention encoder"? Information on the number of layers, attention heads, and total parameter count for this module would be very helpful for reproducibility.
3.  The current similarity metric for example retrieval is based on L2 distance. Have the authors explored or considered more semantic similarity metrics for retrieving relevant experiences, which might be more robust in complex scenarios where simple coordinate distance may not fully capture contextual similarity?
4.  The reliance on regular expressions for output parsing is a potential point of failure. Have the authors measured the parsing failure rate in their experiments? Have they considered more robust alternatives, such as prompting the LLM to generate a structured format like JSON or using constrained decoding, to ensure output reliability?

---

### Official Review · Reviewer_JxvX · 2025-11-01

**Soundness:** 2
**Presentation:** 2
**Contribution:** 2
**Rating:** 4
**Confidence:** 3

**Summary:**

### Summary

This paper propose LLM-TrajEcho framework, which unifies a lightweight vision encoder, the LLM reasoning and online-feedback in end-to-end trajectory predcition task. This approach aims to achieve high accuracy and real-time performance without the need for expensive sensor fusion and fully leverage the contextural reasoning capacity of LLM.

**Strengths:**

Strengths:

- The paper demonstrates excellent results across multiple benchmarks such as nuScenes, KITTI, and Waymo. Achieving a state-of-the-art minFDE of 2.45 on nuScenes and the lowest Miss Rates (MR) provides strong quantitative evidence for the framework's effectiveness.
- The framework is designed for real-time performance, which is a critical requirement for practical deployment. The inference time comparisons show a significant advantage compared to previous methods.
- The extensive experiments, including comprehensive ablation studies in Table.4 and Table.8, systematically validate the contribution of each component in LLM-TrajEcho.

**Weaknesses:**

Weaknesses:

- The visual encoder is kept frozen during training to ensure stability and this makes the entire system's performance highly dependent on the quality and capabilities of the pre-selected encoder. An investigation into the impact of different vision backbones would have strengthened the paper. Moreover, the lightweight vision encoder including patch-wise self-attention lacks novelty and this paper lacks of the inference speed with different *P* in the vision encoder which is an important factor.
- The authors claim that LLM-TrajEcho is the first method that uses structured natural language as the motion representation. However, the *Language-as-Motion-Representation* is not novel. Such as in OmniDrive[1], the language-style motion representation is already used to instruct the LLM-agent to reason the planning behaviors. Author need to explain the innovation of motion representation in LLM-TrajEcho, given that the author emphasizes that this method is the first attempt to use this approach.
- Doubts about the effectiveness of the modules proposed in LLM-TrajEcho. In Table 4, the author demonstrates the effectiveness of three modules in LLM TrajEcho. However, the impact of any module is limited. The significant improvement in minFDE metrics (5.41->2.45) does not come from these modules. Based on all experiments, does the improvement in minFDE come from the frozen vision encoder?  However嚗蘏s stated in Weakness#1, the authors did not specify what vision encoder was used and what impact such a vision encoder has.
- Ambiguity of Online Feedback module. An important contribution of LLM-TrajEcho is the proposal of online updating contextual examples. However, in section 4.3, the authors did not specify what *s* and *a* are, and *ADE (s, a)* does not correspond to any part in Equation 8.
- The significant improvement in terms of time consumption is worthy of detailed explanation. In LLM-TrajEcho, there are video vision encoder and LLM inference parts, both of which contain a huge number of parameters, resulting in high latency. It is strongly recommended that the author explain the image resolution, the patch size, the number of vision encoder parameters, and the inference time required for each module.
- Minor Weakness: The author's visualization in Figure 2 is blurry and the legend is obscured. The author needs to optimize these qualitative result.

[1] Wang, Shihao, et al. "Omnidrive: A holistic llm-agent framework for autonomous driving with 3d perception, reasoning and planning." CoRR (2024).

**Questions:**

na

---

### Official Review · Reviewer_9sgd · 2025-11-11

**Soundness:** 3
**Presentation:** 2
**Contribution:** 2
**Rating:** 4
**Confidence:** 3

**Summary:**

- The paper introduces a lightweight end-to-end framework that encodes video features and fine-tunes an LLM to predict future trajectories.
- It uses a closed-loop feedback mechanism that updates in-context examples based on online prediction errors.

**Strengths:**

- It shows the effectiveness of combining a pretrained vision encoder with a properly finetuned LLM, to quickly and accurately predict trajectories directly from videos.
- It achieves competitive results in nuScene and KITTI.

**Weaknesses:**

1. It is not entirely clear how the closed-loop feedback mechanism operates during inference, especially when ground truth future trajectories are unavailable. What does the template look like?
2. For the comparison with baselines, it would be helpful to categorize them into those that use only historical poses, only videos, or both. (See more in the questions section.)
3. The overview figure (Fig. 1) could be improved — it does not clearly show how the LLM is trained, and the template design remains vague.

**Questions:**

1. Has the paper compared against baselines that also use video as input? It seems that many of the recent baselines compared in the paper are post-perception models that use only ground-truth bounding boxes as input. The paper seems to use both ground-truth historical positions (in the prompt) and video inputs. Are there baselines with a similar setup? If so, please compare with them.
2. Regarding the metrics for nuScenes (e.g., minFDE1, minADE5, minADE10), could you elaborate on how control the LLM to generate multiple trajectory samples using the posterior estimate in Eq. (7)?
3. What is the loss function of LLM? How does it link with the online feedback mechanism's reward function?

---

### Note · Authors · 2025-11-14

I have read and agree with the venue's withdrawal policy on behalf of myself and my co-authors.